# Cross-Layer and SDN Based Routing Scheme for P2P Communication in Vehicular Ad-Hoc Networks

**Ziyi You [1,2,*], Guixian Cheng [1,2], Yi Wang [2,3], Pu Chen [1,2] and Shiguo Chen [1]**

[1] Department of Physics & Electronic Science, Guizhou Normal University, Guiyang 550001, China; chgx86@126.com (G.C.); 18085122220@189.com (P.C.); 18184116660@189.com (S.C.)
[2] Key Laboratory of special Automotive Electronics technology of the Education Department of Guizhou Province, Guiyang, Guizhou 550001, China; 13007888851@126.com
[3] Guizhou City Vocational College, Guiyang 550025, China
[*] Correspondence: 100232632@gznu.edu.cn; Tel.: +86-13765179210

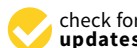

**Featured Application: Authors are encouraged to provide a concise description of the specific application or a potential application of the work. This section is not mandatory.**

**Abstract:** Conventional routing protocols proposed for Vehicular Ad-hoc Network (VANET) are usually inefficient and vulnerable for multi-hop data forwarding due to the unavailability of global information and inefficiencies in their route discovering schemes. However, with the recently emerged software defined vehicular network (SDVN) technologies, link stability can be better improved through the availability of global network information. Thus, in this paper, we present a novel software-defined network (SDN) based routing scheme for P2P connection under urban inter-vehicle networks that can find a global optimal route between source and destination. This is a cross-layer routing protocol in VANETs, which utilizes metrics not only considering the position and velocity of vehicles, but also channel allocation and link duration when selecting the relay vehicles. Consequently, it starts a route discovery process which can improve the network performance in terms of end-to-end delay and low overhead. Furthermore, packet loss is largely minimized by the relatively stable paths. With the help of realistic simulation, we show that the proposed routing framework performs better than other three latest SDVN and conventional VANET protocols in routing overhead, average end-to-end delay, packet drop ratio, and average throughput. Therefore, our routing scheme is more suitable for 5G-enabled vehicular ad-hoc networks in future.

**Keywords:** software-defined networking (SDN); inter-vehicle network; data routing; cross-layer design; performance analysis

## 1. Introduction

Vehicular Ad-hoc Network (VANET), as one of the core technologies of the Intelligent Transportation System (ITS), has huge application potential and commercial value. Based on the Vehicle-to-Vehicle (V2V) communications and the vehicle-to-roadside communications (V2R) [1,2], VANETs are aimed at the applications including traffic information dissemination, traffic safety, and Internet access services, etc. Furthermore, with the emergence of the Internet of everything in 5G [3,4], VANET has been given a new functional role, as a natural extension of the mobile network in the road travel scenario.

The inherent characteristics present in VANETs i.e., the diverse node velocities, severe channel fading, dynamic network topology, and limit transmission coverage of vehicles, etc., impose a lot of challenges in designing an efficient routing algorithm with both short delivery delay time and

high reliability. Conventional routing protocols in VANETs are usually topology-based routing or geographical-based routing. The topology-based protocols (e.g., Ad hoc On-Demand Distance Vector Routing (AODV) and Dynamic Source Routing(DSR)) which utilize the link information in the networks are not suited for vehicular ad hoc networks because links between vehicles can break down so frequently that it is hard to discover a multi-hop route from the source to the destination [5]. Even though the authors in [6] have proposed an efficient AODV with backbone based routing for message dissemination to increase packet delivery ratio, the performance of this protocol is still limited during low vehicle density due to frequent link breaks. In addition, geographic-based protocols (e.g., Greedy Perimeter Stateless Routing (GPSR), Geographic Source Routing (GSR)and Geographical and Energy Aware Routing (GEAR) [7,8] only need local neighborhood information to perform packet forwarding and require quite low routing overhead, but this type of routing can lead to local optimization and causes a long end-to-end packet delay. All these routing protocols above used in traditional networks can hardly meet the requirements of efficient data exchanging on internet of vehicles (IOVs), such as data transmission efficiency, network resource utilization rate, etc. Software-defined network (SDN) [9,10], as a model of the next generation network, could be rationally adopted for the data exchanging network of IOVs, and it will bring some benefits of introducing SDN technology to VANET. In SDN-based VANET, it is not necessary for vehicles to exchange routing messages with each other which causes the routing overhead and end-to-end packet delay to be greatly improved.

So far, researchers have investigated more how to take advantage of SDN benefits to improve the performance of current VANETs. Some existing works [11–13] propose fully SDN-based architectures to extend SDN to operate in wireless vehicular network scenarios. The resulting Software Defined Vehicular Network (SDVN) provides a bird's eye view over the network, and thus, the link stability can be better scrutinized. The authors in [14] propose a SDVN architecture and relative services, they also demonstrate in simulation the advantage of a SDVN routing on the reliability, flexibility, and programmability. The authors in [15] present a SDN-based geographic routing (SDGR) protocol to address the local maximum problem in sparse networks. The controller computes the shortest connected path with probability of high connectivity, and with this, a packet forwarding algorithm is proposed to select the next hop to solve the load balance problem at junctions. SDN-based vehicular ad-hoc on-demand (SVAO) routing protocol [16] separates the control layer and the data forwarding layer to improve network performance in VANETs. The protocol is an amendment of the ad hoc on-demand distance vector (AODV) protocol, in which both global and local controllers are used to improve the packet reception rate. In addition, it is feasible and important for combining vehicles with cloud computing for the development of IOV, which will solve the communication decencies problem of the vehicle-to-everything (V2X) [17,18]. V2X is a form of technology that allows vehicles to communicate with moving parts of the traffic system around them, including V2V and V2I.

None of the routing protocols described above consider combing the wireless channel access constraint with the mobility associated with links to compute an optimized route. Our goal is to present an efficient cross-layer SDVN routing solution for message propagation based on previous works [11,14,15,17]. The proposed protocol selects appropriate relay nodes by calculating several parameters including vehicle velocity, node position information, link stability, and wireless bandwidth allocation. The following are the key contributions of this paper.

(i) An in-depth discussion about SDVN is described in terms of its main components and the benefits of introducing SDN technologies to vehicular networks.

(ii) A novel cross-layer SDVN routing protocol is proposed to find a more robust route between source and destination. We intend to solve the problems of spectrum scarcity, intermittently connected networks, and high latency. The beauty of the proposed scheme is not only the efficiency of the route discovery process, but also the quality of the available paths.

The rest of the paper is organized as follows. The existing SDVN architecture is described in Section 2.1. The proposed routing protocol is described in detail in Section 2.2 and followed by a performance analysis and simulation result in Section 3. Finally, in Section 4, the conclusion is given.

## 2. Methods

### *2.1. SDVN Framework*

In order to implement SDN to VANET, the architecture of SDN based routing framework is presented in [19,20]. As shown in Figure 1, the whole vehicular network consists of three types of components, i.e., SDN controller, local controller, and forwarding nodes. Furthermore, each road is divided into several segments of equal length. The SDN controller is responsible for keeping global, updated information about the network topology, and it helps the network to provide the best stable route by maximizing the path duration among all the paths between source and destination. Those Road Side Units (RSUs) that serve as local controllers reduce the burden of the SDN controllers by keeping local, updated information about the network topology within communications range. Notice that the local controller is only responsible for sending control information to mobile vehicles within its coverage, but it does not need to deliver data information to vehicles. We assume that the RSUs serve as gateways on the VANET part thereby connecting with traditional clouds. Hence, local controllers cooperate with the SDN controllers on clouds to maintain the global network topology. The OpenFlow protocol is adapted for communication between the data plane and the control plane, the key benefit of which is that the existing hardware can be efficiently utilized under SDN [21].

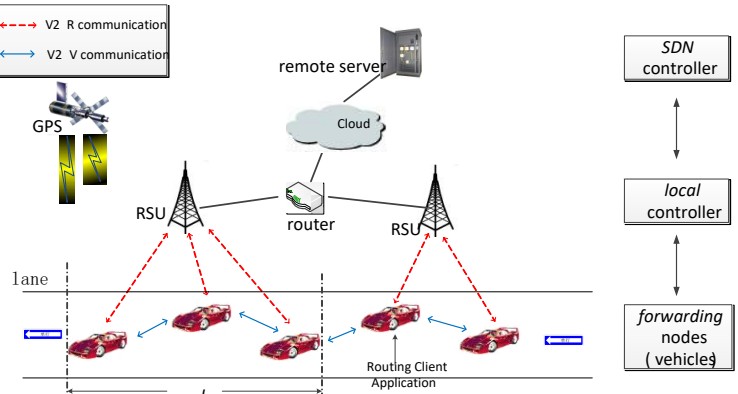

**Figure 1.** The architecture of software-defined network (SDN)-based routing framework.

### 2.1.1. Registering Phase

In this framework, each vehicle uses hello messages, including its velocity, location, pseudo-ID, and channel state information (CSI) etc. to periodically advertise status information to the nearest local controller at a fixed interval. After the corresponding controller receives these "State Update" messages, it will update the vehicles' state information tables. Note that each local controller separately maintains a vehicular history trace database which records the state information of all vehicles in its control area, and all vehicles' state information from these local databases could be integrated into a global network state vector via cloud computing.

### 2.1.2. Route Prediction Phase

In the P2P communication process, the source node firstly needs to send a route request (RREQ) packet to the nearest RSU. The RREQ message contains the source's ID and destination's ID. When receiving RREQ, the routing server will calculate a more reliable and optimized routing path from source to destination using global network state vector and our proposed routing algorithm. After that, this routing server will respectively respond a route reply (RREP) to all nodes on the routing

path (including source, destination, and other intermediate vehicles) via local controllers. When these nodes receive the respective RREP message, they insert the routing path mark and next hop node ID to their own routing tables. Then the data flow will be sent out and propagated along the route to the destination. The route will stay alive for a certain length of time, after this time, it will be deleted so that the new routing path will be recomputed for the next cycle. This alive-timeout length is important to make our routing algorithm adaptive to the dynamic network topology. In addition, the routing path also will be limited by a certain maximum number of hops, so that the communication distance between source and destination will not be too long to reduce the efficiency of data packet transmission. The route discovery process in detail is described in Algorithm 1, and the whole session process of our protocol in detail is described in Algorithm 2.

### 2.1.3. Access Wireless Communication Protocol

SDN technology in VANETs also allow using multiple wireless technologies, like DSRC (Dedicated Short Range Communications), WAVE (Wireless Access in the Vehicular Environment), and LTE (Long Term Evaluation), etc. The relationship between cellular and ad-hoc communications technologies is suggested to be complementary; hence it will be a promising scheme to extend LTE with the direct communication capability between vehicles to become an integrated V2X solution. In this paper, we assume that the proposed algorithm adopts an LTE-A (LTE-Advanced) network [22,23] for communication between vehicle and vehicle.

### *2.2. Proposed Routing Scheme*

### 2.2.1. Routing Metrics

Cross layer design has been widely used to improve the network performance in wireless networks [24]. Such networks are expected to support various types of applications with different and multiple QoS and grade-of-service (GoS) requirements. Our proposed algorithm exploits cross-layer information exchange to obtain the most optimal routing path, which depends on three different parameters, i.e., forwarding probability, link duration, and wireless bandwidth allocation.

*A.   Forwarding probability*

The forwarding probability of any intermediate node could be calculated [25] according to the following formula.

$$P_{forwarding} = \alpha_1 \frac{V}{V_{max}} + \alpha_2 \frac{D}{R},$$ (1)

where the constants $\alpha_1$ and $\alpha_2$ are used to weight the contribution of both sub-objectives, and $\alpha_1 + \alpha_2 = 1$. V Denotes the velocity of a current intermediate node, and $V_{max}$ denotes the fastest velocity allowed on this road section. $D$ is the calculating distance between the previous forwarding node and the current intermediate node. $R$ denotes the communication range between vehicles which is a fixed value.

However, the main application scenario in [25] is expressway. Compared with the expressway scenarios, the urban road topology is complex and changeable. In urban scenarios, the communication range of vehicles is affected by many factors, and there are differences in the communication range between roads and intersections. Therefore, it is not appropriate to use fixed $R$ to express the communication range. In this paper we derive adaptive transmission range between vehicles by the knowledge about the velocity and arrival rate. Then Equation (1) is modified to Equation (2).

$$P_{frowarding} = \alpha_1 \frac{V}{V_{max}} + \alpha_2 \frac{D}{\min\{L_i(1 - K_{ei})\sqrt{\frac{L_i \ln L_i}{K_{ei}}} + \beta L\}},$$ (2)

the normalized vehicle density on road segment *i*.
where $\beta \in (0, 1)$ denotes traffic flow constant [26]. $L_i$ denotes the length of road segment *i*. $K_{ei}$ denotes
*B.   Link duration*

The communication link lifetime between two adjacent mobile nodes $i$ and $j(CLT_{i,j})$ on the route is calculated as Equation (3):

$$CLT_{i,j} = \frac{-(ab + cd) + \sqrt{(a^2 + c^2)R^2 - (ad - bc)^2}}{a^2 + c^2}. \tag{3}$$

In Equation (3) $R$ is the communication radius between node $i$ and node $j$. $a$, $b$, $c$ and $d$ are separately defined as follows.

$$a = v_i \cos\theta_i - v_j \cos\theta_j$$
$$b = x_i - x_j$$
$$c = v_i \sin\theta_i - v_j \sin\theta_j$$
$$d = y_i - y_j,$$

where $v_i$ and $v_j$ separately denote the velocity of $i$ and $j$. $(x_i, y_i)$ and $(x_j, y_j)$ separately denote the position of $i$ and $j$ which can be obtained by GPS. $\theta_i$ and $\theta_j$ separately denote the velocity angle of $i$ and $j$.

Among all the neighboring vehicles within communication range of the transmitting node $i$, the one that has the maximum $CLT_{i,j}$ ($j \in$ neigbor of $i$) can be selected first. According to Equation. (3) above, the link duration is determined by each $CLT_{i,i+1}$ of Node $i$ $(1 < i < N - 1)$ on the route as Equation (4).

$$lifetime = \min\{CLT_{1,2}, CLT_{2,3}, \dots, CLT_{N-1,N}\}, \tag{4}$$

where $N$ denotes number of hops of the route and $N$ must be less than certain threshold $k$. It means that the path length in the proposed routing algorithm is limited to a short distance. Similarly, *lifetime* must be greater than or equal to a specified time interval in order to ensure the link stability of our algorithm.

### C.   *Wireless bandwidth allocation*

Bandwidth estimation is necessary to transmit the data, because nodes within the transmission and interference range of each other will consume the bandwidth of each other. Suppose the LTE-A system is used in this inter-vehicle network architecture, the available wireless bandwidth between any two adjacent mobile nodes ($i$ and $j$) may be figured out using following formula.

$$BW_{i,j} = B_i \cdot \log_2(1 + \frac{p_i G_{i,j}}{N_0 + p_c G_{c,j}}), \tag{5}$$

where $p_i$ denotes the transmitting power of sender $i$, $p_c$ denotes the transmitting power of the cellular user c, which will allocate a portion of channel bandwidth to the data transmission between $i$ and $j$. $B_i$ denotes channel bandwidth of c. $G_{i,j}$ denotes the link gain between sender $i$ and receiver $j$. $G_{c,j}$ denotes the link gain between c and $j$. $N_0$ is the noise power.

Obviously, $BW_{i,j}$ must reach the level of corresponding application business, otherwise, the route building will not succeed.

### 2.2.2. Routing Format

In Section 2, we mentioned that local controllers need to create a RREP packet respectively for each node along the route and unicast it to the corresponding node in order to establish one link. Furthermore, the RREP packets will be updated and transmitted periodically to maintain the link until the end of data transmission. The RREP format consists of fields as shown in Figure 2, where *Rsu_ID* denotes the RSU number. *Vehicle_ ID* denotes the vehicle identity of destination. *Package ID* denotes the RREP packet number. *Last Hop* field denotes the vehicle identity of the previous hop on the route, and *Next Hop* denotes vehicle identity of the next hop on the route. *SendTime* is the sending time of the RREP packet. *DelayTime* is the delay time when the destination receives this routing packet. *ExpiredTime* denotes the expiration time of this routing packet. *CMC* is the specific check

code encrypted by the RSU's private key that verifies the correctness of the packet. As can be seen in Figure 2, the length of RREP packet in the proposed algorithm is shorter than the AODV algorithm, and the local controllers only need to send a RREP packet to each vehicle node on the route. Therefore, we believe that our algorithm performs better in packet overhead, compared to AODV.

| Rsu_ID | Vehicle_ID |
|---|---|
| Package_ID | |
| Last Hop | Next Hop |
| SendTime | DelayTime |
| ExpiredTime | CMC |

**Figure 2.** Route reply (RREP) packet.

Each vehicle stores a simple routing table inside. When receiving the RREP packet, these nodes need to update the information of their own routing table immediately. Note that the link duration should be greater than or equal to the sending cycle of the RREP packet.

### 2.2.3. Routing Table

The proposed algorithm provides that all the vehicle nodes in VANET need to store a simple routing table. Once the route is selected, each node along the path creates a temporary forwarding table which specifies the path for forwarding the packets. In the forwarding table, the field *Next hop* denotes the next hop on the direction from source node *S* to destination node *D*. *Last hop* denotes the next hop on the direction from *D* to *S*. *ID* denotes the ID number of links between *D* and *S*. *Destination sequence number* denotes the RREP sequence number.

### 2.2.4. Route Establishment and Data Transmission

Our algorithm adopts the cross-layer computation principle for the route establishment process. Depending on the vehicle's velocity and position, wireless bandwidth allocation and longest link duration, an optimized path is chosen by the local controllers cooperating with SDN controllers for data flow transmission.

Algorithm 1 shows the detail of the route discovery in our algorithm. These cross-layer metrics mentioned above are used to establish the routing path from source vehicle to destination vehicle, as shown in lines 12–14. Note that, the distance between source vehicle and destination vehicle is limited (lines 5 and 6), and the stack structure 'PathSet' could be used to store the route (lines 19–23).

Algorithm 2 describes the whole session process of the proposed SDVN routing protocol, which can be divided into following four steps:

(i)　each vehicle sends "state updating" messages (line 2),
(ii)　the source vehicle sends RREQ packet (line 3),
(iii)　the local controllers distribute RREP packets for link establishment (lines 4–10),
(iv)　route refreshing phase (lines 11–14).

When link damage is detected (line 11), check whether the P2P path can be repaired. If it can be repaired, continue to perform the data transmission along the repaired path. For instance, there is a vehicle $v_{new}$ that can replace the broken node $v_i$ while other nodes on the routing path have not changed, then the repaired path is $s <-> v_1 <-> \ldots <-> v_{i-1} <-> v_{new} <-> v_{i+1} <-> \ldots <-> v_n <-> d$. If the P2P path cannot be repaired, restart the routing discovery process to find another new P2P path. Figure 3 shows how the source selects the local controller and communicates with the SDN controller to find the final route from among all the paths between source and destination.

---

**Algorithm 1** route discovery process.

---

**Input**: NodeSet
**Output:** Routing path from source vehicle to destination
**Initialize**: CycleSet, PathSet, Head, Tail
**Notation:**
NodeSet: The vehicle's location at current cycle
CycleSet: The prediction of vehicle's location at next cycle
PathSet: The stack used to store routing path
1. **Begin**
2. if NodeSet is empty
3.    say 'error' and go to 25
4. end if
5. if Distance (Source_node, Destination_ node)
   >Distance_Threshold
6.    say 'error' and go to 25
7. CycleSet = calculate NodeSet location after
   Cycle
8. Head = Source_node
9. Tail = Destination_node
10. Push (PathSet, Head)
11. Sort CycleSet by Distance (CycleSet, Head)
12. for each $Node_i \in$ CycleSet from Head to Tail
13.    if Distance (Head, $Node_i.(x, y))$ <
       Communication Radius of Vehicle AND
       $Node_i.P_{forwarding}$ > threshold value $P'$
14.       if Available_bandwith $BW_{Nodei,head}$ > threshold
          $BW'$ AND $CLT_{i,Head}$ > interval time $T'$
15.          Push (PathSet, $Node_i$) AND
             Head = $Node_i$
16.       end if
17.    end if
18. end for
19. check if PathSet.top! = Tail AND
    PathSet.top! = Null
20.    $Node_x$ = pop(PathSet) AND delete $Node_x$
       from CycleSet AND Head = PathSet.top
       AND go to 12
21. end if
22. if PathSet.top = Null
       say' error' and go to 25
23. end if
24. say 'success'
25. **END Algorithm 1**

---

---

**Algorithm 2** the whole session process.

---

**Input:** Source vehicle $v$
**Output:** Forward the data packet toward destination vehicle
**Notation:**
PathSet: The data structure used to store routing path
CSI: The channel state information
Next_Hop: The next hop vehicle
1. **Begin**
2. Each vehicle periodically unicasts message
    "*Hello_msg*" to nearest RSU with fields: velocity, location, pseudo-ID, CSI
3. The source vehicle $s$ generates data to send
    AND sends RREQ message to nearest RSU
4. The local controllers cooperate with SDN
    controllers establish Pathset by Algorithm 1
    OR repair the damaged path
5. The controller group $C$ sets an interval
    Time $T$
6. for each controller $c_j \in C$
        $c_j$ unicasts RREPi message to $Node_i$ within its coverage AND $Node_i \in$ Pathset with fields: Rsu_ID,
Vehicle_ID, Package_ID, Last_Hop, Next_Hop, SendTime, DelayTime, ExpiredTime, CMC
7. end for
8. for each $Node_i \in$ Pathset
9.         when $Node_{i\ receives}$ $RREP_i$ then
        immediately updates the Routing table AND connects with Next_Hop marked in $RREP_i$
10. end for
11. if link establishment fails OR link damages
12.         go to 4
13. else
14.         link maintains Until $T$ time out then go
        to 4 OR data transmission finishes then go to 15
15. end if
16. **END Algorithm 2**

---

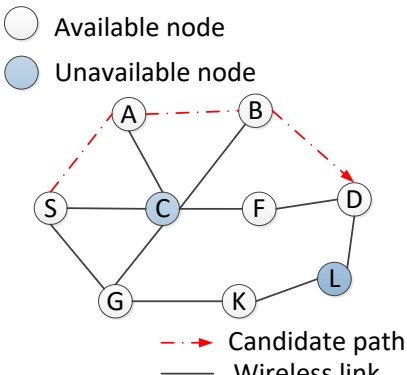

**Figure 3.** Optimal path selection.

## 3. Results and Discussion

### 3.1. Performance Analysis

In this section, we provide a performance estimate of our routing algorithm by analyzing its route selection, link stability, transmission delay etc.

### 3.1.1. Benefits of SDN Migration

The SDN migration may maintain a global vision of the VANET topology. It means that SDN controllers are able to monitor traffic flows in the VANET domain in real time. When data traffic becomes unbalanced, they can start a reroute traffic process to improve network utility and reduce congestion. In addition, the SDN/OpenFlow technology can increase network intelligence by decoupling the control plane from the data plane; the decoupling brings efficiency, flexibility, management. Hence, SDN/OpenFlow allows the system to make more informed routing decisions in terms of path selection, channel selection, and safety service.

### 3.1.2. Cross-Layer Solution

The proposed scheme is designed on cross-layer metrics from MAC layer to application layer, which selects the best route by achieving high probability and low time delay as well as channel availability. For instance, a source vehicle $S$ wants to establish a reliable route to the destination vehicle $D$. In terms of the prediction of topology map in Figure 4, the flows from $S$ to $D$ can be split into multiple paths ($S{\rightarrow}C{\rightarrow}F{\rightarrow}D$, $S{\rightarrow}A{\rightarrow}B{\rightarrow}D$, $S{\rightarrow}G{\rightarrow}K{\rightarrow}L{\rightarrow}D$ and $S{\rightarrow}G{\rightarrow}C{\rightarrow}F{\rightarrow}D$). Suppose Vehicle $C$'s available bandwidth is less than the given threshold value that means it has not enough channel bandwidth to forward data, so the paths, i.e., $S{\rightarrow}C{\rightarrow}F{\rightarrow}D$ and $S{\rightarrow}G{\rightarrow}C{\rightarrow}F{\rightarrow}D$ are not available for the specified business type. In addition, vehicle $L$'s reception probability in path $G{\rightarrow}K{\rightarrow}L{\rightarrow}D$ is less than the threshold probability, hence, this path also cannot be used. Finally, the path $S{\rightarrow}A{\rightarrow}B{\rightarrow}D$ is selected as the reliable path at the next cycle.

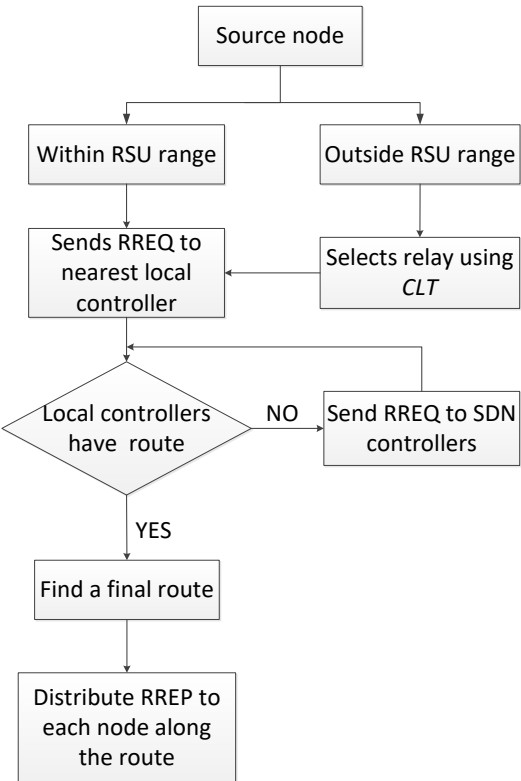

**Figure 4.** A flowchart representing two cases of local controllers finding a final route.

If the Dijkstra algorithm is used in Figure 3, the shortest path could be $S{\rightarrow}C{\rightarrow}f{\rightarrow}D$ in terms of inter-vehicle distance. However, selecting this path will lead to link quality degradation or even link damages since the available bandwidth of node C is not enough. Therefore, compared to the Dijkstra algorithm, our algorithm performs better.

### 3.1.3. Communication Load

In order to maintain the state of the link between each pair of nodes, all control packets are periodically distributed by local controllers, and then a routing table is constructed at each node on the routing path. This method makes the source vehicle not broadcast a routing request message to neighbor vehicles like AODV protocol, which will alleviate the multicast traffic flooding problem in the routing discovery process to a large extent.

### 3.2. Simulation Evaluation

In this section, we use OPNET14.5 to analyze the performance of our routing algorithm, and compare it with efficient AODV without SDN [6], SVAO with SDN [24], and SDGR with SDN [23] in the same urban scenario.

The simulations were carried out considering a roadway field having a $10 \times 5$ km rectangular shape, and the running time was set to 5 min. As shown in Figure 5, there are three 4 km $\times$ 100 m one-way lanes named A, B and C, and assuming that A and B are in opposite directions while C is perpendicular to A and B. There are four RSUs placed near four corners of the rectangular simulation area, which ensures that all vehicles in the area are within the coverage of SDN network. To estimate the impact of density on the performance of the routing protocols, we varied the number of nodes from 15, 30, 50, and above 70 nodes. The network environment parameters are shown in Table 1.

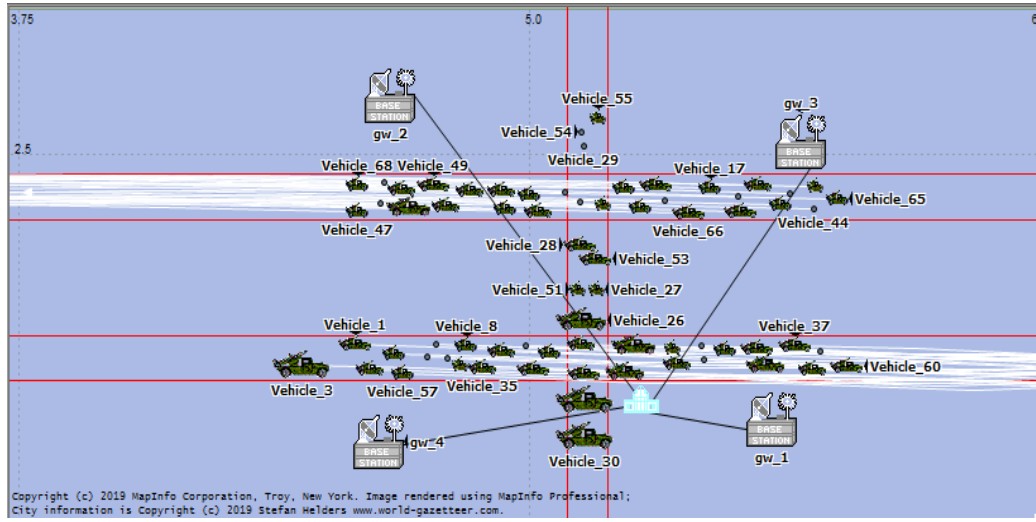

**Figure 5.** Simulation scenario.

**Table 1.** Simulation parameters.

| Number of Nodes | 15, 30, 50, above 70 |
| --- | --- |
| Total simulation time | 300 s |
| Area size | $10 \times 5$ km |
| Packet size | 1024 bytes |
| MAC protocol | IEEE 802.11 p |
| Mobility Model | Random Waypoint |
| Vehicle moving speed | 15–20 km/h |
| Vehicle max speed | 25 km/h |
| Radio range | 500 m |
| Link alive time | 5 s |
| Broadcasting cycle | 1–1.5 s |
| Max distance | 2 km |
| Protocols | Efficient AODV, SVAO, SDGR, the proposed algorithm |

The performance metrics that we considered were routing overhead, average end-to-end delay, packet drop ratio, and average throughput.

*Routing overhead*: Ratio of the routing packets generated to the total data packets delivered to the receivers.

*Average end-to-end delay*: Average delay experienced by a data packet across the inter-vehicle network from one source vehicle to another destination.

*Packet drop ratio*: Ratio of the number of data packets lost at the destinations to the number of data packets generated by the source vehicle.

*Average throughput*: Average rate of successful packets (including routing packets and data packets) delivery over the inter-vehicle network.

Figure 6 shows the performance comparison of routing overhead for efficient AODV, SVAO, and SDGR and the proposed algorithm. From the figure, it can be seen that the proposed scheme outperforms the other three reference schemes, consistently maintains around 0.21 in all scenarios: 15, 30, 50 and above 70 nodes. The SDN controllers keep a global record of updated information due to the cooperation among the local controllers and vehicles. The selection of local controllers at different depths from the SDN controllers reduces unnecessary propagation of control messages; thereby the overhead ratio of SDN-based schemes shows better performance than the non-SDN-based approach i.e., efficient AODV. On the other hand, the distance-based selections in the reference protocols SVAO and SDGR, show performance degradation in terms of overhead ratio, as the stability is not assured.

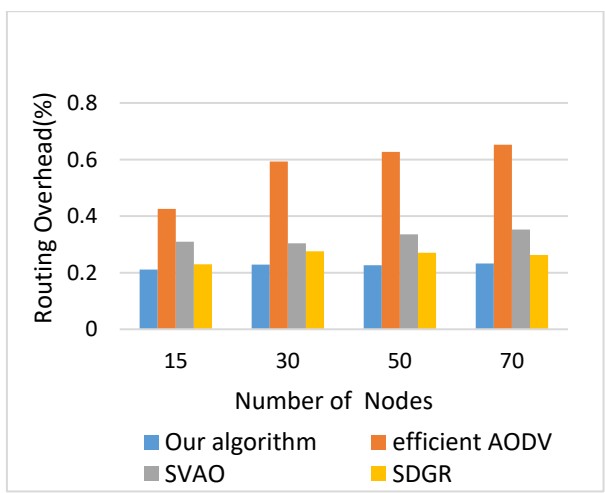

**Figure 6.** Routing overhead comparison.

Figure 7 illustrates the comparison between average end-to-end delay as a function of the number of vehicles. As can be seen from the figure, with an increasing number of vehicular nodes in the network, the end-to-end delay decreases. The pattern is the same for all the schemes, since the connectivity in the network will increase with an increase in vehicular nodes. Obviously, SDN-based protocols show much less end-to-end delay than the efficient AODV because applying the local controllers can reduce the network delay. In addition, the SVAO and SDGR select the next node based on distance only, whereas the proposed scheme calculates metrics including forwarding probability, link duration, and wireless bandwidth allocation. Hence, our target is to maintain network stability by providing the optimized route between source and destination, as a result, the proposed scheme shows less end-to-end delay than other two SDN-based schemes.

As shown in Figure 8, we can see that the proposed algorithm is also the best for packet drop ratio under the assumed simulation scenario, whose packet drop ratio is almost 0% until the number of 50 nodes. In the efficient AODV without SDN, a querying node has to select the next node for every hop until it reaches the destination. By doing so, it may come across several fragile links due to the random and fast movement of vehicles. However, in other SDN-based schemes, the SDN controllers

keep the global view of the VANET, which means the SDN controllers manage all the information about idle channels and relay nodes for each segment of the road. Therefore, SDN controllers can provide more informed routing decisions to each querying node. Furthermore, our proposed protocol increases network stability by providing a more stable route solution, for this reason, the least number of packets dropped compared to the other three protocols. Note that, the result above is obtained in the simulation environment we set, not in all simulation scenarios, and we did not take into account other factors not mentioned in this paper [27].

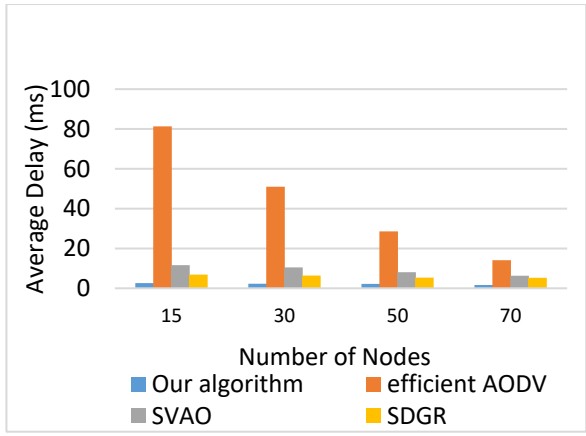

**Figure 7.** Average end-to-end delay.

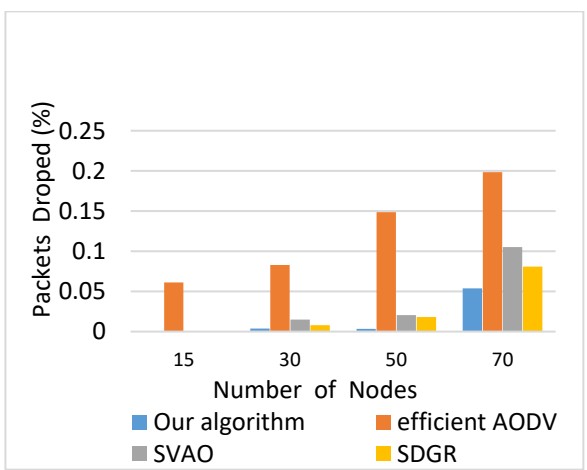

**Figure 8.** Packet drop ratio.

Figure 9 shows the impact of the proposed scheme on average throughput. It can be clearly observed that as number of vehicles increases, the throughput of our proposed algorithm is relatively stable and higher than efficient AODV, SVAO, and SDGR. Specifically, the throughput of our algorithm is in the range of 25,000–30,000 bit/s for the considered range of vehicle nodes (50–above 70). This is because the proposed algorithm is aware of stable links and the future position of nodes, which result in lower packet loss. Furthermore, it does not consume more bandwidth and gives opportunity to other packets to transmit in the inter-vehicle network. Obviously, our algorithm can receive a greater number of data packets than the other three protocols.

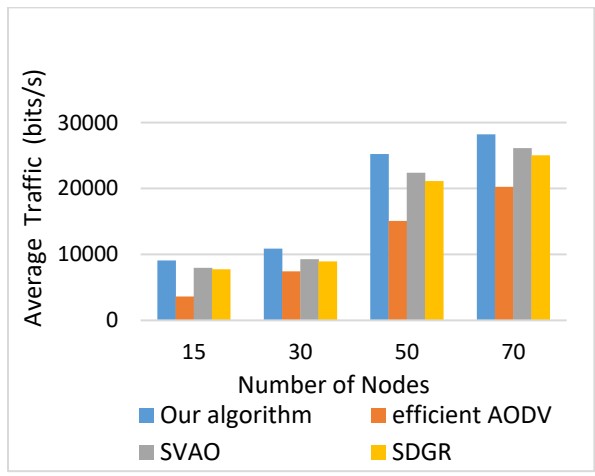

**Figure 9.** Average throughput.

## 4. Conclusions

In this paper, an efficient cross-layer SDVN routing scheme for P2P communication was presented. The novelty of this work lies in its unique design based on Software-Defined vehicular networks. Taking into account the existing SDVN framework, the proposed algorithm calculates globally optimized routes for V2V communication by SDN/OpenFlow controllers, while generating much lower end-to-end delay. Specially, the path selection from source to destination is determined based on the cross-layer metrics including forwarding probability, available bandwidth, and link duration. Consequently, its link stability-based routing decision also reduces the probability of link failure resulting in a lower rate of path disconnection. The overall performance of our proposed scheme was evaluated with varying vehicle speed and density, and compared with efficient AODV without SDN, SVAO with SDN, and SDGR with SDN in simulation scenarios. The result has proven that it is superior to the other three protocols in terms of routing overhead, average end-to-end delay, packet drop ratio, and average throughput.

Fifth generation (5G) VANET is considered as a promising technology due to the merits of 5G architecture such as high mobility and scalability support, massive connectivity, and reduced latency, which overcome the limitations of the 4G technology. The proposed scheme is reliable and efficient, which is suitable for 5G applications. In addition, the SDN-based global routing greatly increases the computing load of remote controllers, which can be further solved by cloud computing and big data technologies.

**Author Contributions:** Methodology, Z.Y. and G.C.; software, P.C.; validation, Z.Y. and P.C.; investigation, Y.W.; writing—original draft preparation, Z.Y.; writing—review and editing, S.C. and G.C.

**Funding:** This research was supported by the National Natural Science Fund under grand 61462015, was also partially funded by the University-Industry-Research Innovation Foundation of Ministry of Education No. 2018A05021 and the Science and Technology Foundation of Guizhou Province No. LH [2016] 7223 and the Science and Technology Foundation of Guizhou Province No. LH [2014] 7045 and the Science and Technology Foundation of Guizhou Province No. LH [2014] 7049.

**Conflicts of Interest:** The authors declare no conflict of interest.

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
