# Peer review of "Cross-Layer and SDN Based Routing Scheme for P2P Communication in Vehicular Ad-Hoc Networks"

_applsci, doi:10.3390/app9224734_

Round 1

Reviewer 1 Report

Dear authors,

The following comments could be useful to improve the quality of your content:

You said at lines 142-143, that the proposed algorithm depends on the three parameters. These parameters are not clearly reflected in Algorithms 1 and 2; Section 3.1, Performance Analysis, has some general information that could not consider in the Results and Analysis;

Figure 3, namely optimal path selection, is Fig. 4, and Fig. 4 is Fig. 5; Figure 4 (Simulation scenario) has poor quality; please, give some hints on why is proper to have the proposed scenario. Remove the title inside the Figs. 6-7 (it is duplicated with the Fig title); Define notations RSU and CLT. Usually, the references are cited in the paper in ascending order; for example, in the line 34 should be [1, 2].

Author Response

Response to Reviewer #1:

Thank you very much for your comments on our paper and the comments have been fair, encouraging and constructive. We have revised the paper according to your comments. The main revisions are listed as follows:

(1)You said at lines 142-143, that the proposed algorithm depends on the three parameters. These parameters are not clearly reflected in Algorithms 1 and 2;

Answer: These three parameters have been added to algorithm 1 at lines 13-14, highlighted in red. In addition, at lines 178-180 some symbols have been modified.

(2)Section 3.1, Performance Analysis, has some general information that could not consider in the Results and Analysis;

Answer: The proposed scheme mainly includes several characteristics such as SDN technology, cross-layer solution and routing discovery process. Therefore, we simply analyze the advantages from three aspects of Benefits of SDN migration, Cross-layer solution and Communication load in Section 3.1. Maybe our analytical process and experimental method are still not deep enough. Thank you for your suggestion, which is very important to us. I have found the shortcomings in my current work, and I will improve the scientific research level according to your suggestions in the future work.

(3)Figure 3, namely optimal path selection, is Fig. 4, and Fig. 4 is Fig. 5;

Answer: Has been modified and highlighted in red.

(4)Figure 4 (Simulation scenario) has poor quality; please, give some hints on why is proper to have the proposed scenario;

Answer: This simulation scenario is easier for us to implement with our existing simulation platform OPNET14.5, in which there are three one-way lanes named A, B, C, and assuming that A and B are in opposite directions. To estimate the impact of density on the performance of the routing protocols, we varied the number of nodes from 15, 30, 50 and 70 above nodes.

Because the communication distance between source and destination is limited by a certain number of hops, we think the simulation scenario used is enough to help us correctly analyze the performance advantages of the proposed method in several aspects e.g. Routing overhead, Average end to end delay, Packet drop ratio and Average throughput. Furthermore, the simulation scenario makes us ignore other complex physical factors not mentioned in this paper and all the routing algorithms for comparison are under the same condition. Thank you for your suggestion, we will improve the simulation method and tool according to your suggestions in the future work.

(5)Remove the title inside the Figs. 6-7 (it is duplicated with the Fig title);

 Answer: Has been modified.

(6)Define notations RSU and CLT.

Answer: Has been modified and highlighted in red.

(7)Usually, the references are cited in the paper in ascending order; for example, in the line 34 should be [1, 2].

Answer: Has been modified and highlighted in red.

Thank you again for your comments, and we have learned much from it. If our revised manuscript has any problem, please don’t hesitate to inform us. Your kind help will be my pleasure.

Reviewer 2 Report

Since on page 10 lines 339-340, the authors wrote that "the proposed algorithm is also the best for packet drop ratio, whose packet drop ratio is almost 0% until the number of 50 nodes". Please demonstrate that by considering the following comments.

In section 3, did the authors consider unidirectional or bidirectional traffic roads? In Figure 4, where are the RSUs places? What is the vehicle mobility model adopted? Please compare Figure 3, page 8 and Dijkstra algorithm used? There are 2 Figure 3, please correct them. Please compare packet overhead (RREP packets) the proposed algorithm and the AODV algorithm.

Author Response

Response to Reviewer #2:

Thank you very much for your comments on our paper. We have carefully considered the comments and have revised the manuscript accordingly. The comments and related answers can be summarized as follows:

Since on page 10 lines 339-340, the authors wrote that "the proposed algorithm is also the best for packet drop ratio, whose packet drop ratio is almost 0% until the number of 50 nodes". Please demonstrate that by considering the following comments.

(1)In section 3, did the authors consider unidirectional or bidirectional traffic roads?

Answer: Yes, we have thought about it. On page 9 lines 311-314, we give the simulation scenario said that there are three one-way lanes named A, B and C, and assuming that A and B are in opposite directions while C is perpendicular to A and B.

(2)In Figure 4, where are the RSUs places?

Answer: There are four RSUs placed near four corners of the rectangular simulation area, which ensures that all vehicles in the area are within the coverage of SDN network. We have added the sentence above at lines 314-316, highlighted in red.

(3)What is the vehicle mobility model adopted?

Answer: Random Waypoint. We have added this parameter into table 2, highlighted in red.

(4)Please compare Figure 3, page 8 and Dijkstra algorithm used?

Answer: If Dijkstra algorithm is used in Figure 3, the shortest path could be S → C → f → D in terms of inter-vehicle distance. However, selecting this path will lead to link quality degradation or even link damages since the available bandwidth of node C is not enough. Therefore, compared to Dijkstra algorithm, our algorithm performs better. We have added the sentence above at lines 292-297, highlighted in red.

(5)There are 2 Figure 3, please correct them.

 Answer: Has been modified and highlighted in red.

(6)Please compare packet overhead (RREP packets) the proposed algorithm and the AODV algorithm.

Answer: As can be seen in Figure 2, the length of RREP packet in the proposed algorithm is shorter than AODV algorithm, and the local controllers only need to send a RREP packet to each vehicle node on the route. So we can believe that our algorithm performs better in packet overhead, compared to AODV. We have added the sentence above at lines 211-214, highlighted in red.

Thank you again for your comments, and we have learned much from it. I will improve the scientific research level according to your suggestions in the future work. If our revised manuscript has any problem, please don’t hesitate to inform us. Your kind help will be my pleasure.

Round 2

Reviewer 1 Report

Dear authors,

I agree with your last adjustments. There are further suggestions:
1. please use a grammar tool to check your paper;
2. define V2X at line 74;
3. in Fig. 1, indicate the local controller; you can write forwarding nodes as italic rather than starting with upper f (see line 94);
4. at line 134, affirmation "et al." is ambiguous for the reader;
5. Table 1 can be removed;
6. The content of Figure 5 must be clear for the reader.

Author Response

Response to Reviewer #1:

Thank you very much for your comments on our paper. We have revised our paper according to your comments as follows.

Please use a grammar tool to check your paper;

Answer: I have checked the grammar of the manuscript with Ginger software and corrected it.

Define V2X at line 74;

Answer: V2X(vehicle-to-everything) is a form of technology that allows  vehicles to communicate with moving parts of the traffic system around them, including V2V and V2I. We have added the sentence above at lines 74-76, highlighted in red.

in Fig. 1, indicate the local controller; you can write forwarding nodes as italic rather than starting with upper f (see line 94);

Answer: Has been modified in Fig.1.

At line 134, affirmation "et al." is ambiguous for the reader;

Answer: At line 134, we have used "etc.” instead of "et al.”, highlighted in red.

Table 1 can be removed;

Answer: Table 1 has been removed and Table 2 becomes Table 1, highlighted in red.

The content of Figure 5 must be clear for the reader.

Answer: The Figure 5 has been remade and shown on the page 9.

       Thank you again for your comments. If our revised manuscript has any problem, please don’t hesitate to inform us. Your kind help will be my pleasure.

Reviewer 2 Report

The VANETs are a rapid change in topology, which has made it really problematic to design a routing protocol for VANETs that is efficient and effective in all terms. The authors should demonstrate sentence on page 10, lines 352-353, "the proposed algorithm is also the best for packet drop ratio, whose packet drop ratio is almost 0% until the number of 50 nodes".

There is not fully to conclude that the proposed algorithm is also the best for the packet drop ratio. Please deeply explain it.

1) The simulation was taken into account at vehicle max speed, 25km/h. How about the movement of the high-speed vehicle?

2) Please refer to this paper.

Muhammad, U. H., Saad Karim, Shah, S. K., Sammar Abbas, Yasin, M., Shahzaib, M., & Umair, M. (2018). A comparative study on frequent link disconnection problems in VANETs. EAI Endorsed Transactions on Energy Web, 5(17) doi:http://dx.doi.org/10.4108/eai.10-4-2018.154444

Author Response

Response to Reviewer #2:

Thank you very much for your comments on our paper. We have revised our paper according to your comments as follows.

The VANETs are a rapid change in topology, which has made it really problematic to design a routing protocol for VANETs that is efficient and effective in all terms. The authors should demonstrate sentence on page 10, lines 352-353, "the proposed algorithm is also the best for packet drop ratio, whose packet drop ratio is almost 0% until the number of 50 nodes".

Answer: This result ,i.e. “the proposed algorithm is also the best for packet drop ratio, whose packet drop ratio is almost 0% until the number of 50 nodes", is obtained in the simulation environment we set, not in all simulation scenarios, and we have not taken into account other factors not mentioned in this paper. So, to make this clear, we change the above sentence to “the proposed algorithm is also the best for packet drop ratio under the assumed simulation scenario, whose packet drop ratio is almost 0% until the number of 50 nodes", highlighted in red.

There is not fully to conclude that the proposed algorithm is also the best for the packet drop ratio. Please deeply explain it.

1.The simulation was taken into account at vehicle max speed, 25km/h. How about the movement of the high-speed vehicle?

Answer: We consider the relative speed between vehicles, rather than the absolute speed of vehicles, because the simulation scenario we give is not suitable for high-speed moving vehicles. But one thing is certain, if the relative speed between vehicles increases, the connection stability will be damaged. Thank you for your suggestion, we will improve the simulation method and tool according to your suggestions in the future work.

2.Please refer to this paper.

Muhammad, U. H., Saad Karim, Shah, S. K., Sammar Abbas, Yasin, M., Shahzaib, M., & Umair, M. (2018). A comparative study on frequent link disconnection problems in VANETs. EAI Endorsed Transactions on Energy Web, 5(17) doi:http://dx.doi.org/10.4108/eai.10-4-2018.154444

Answer: The paper above have been referred in my paper, at lines 369-371, highlighted in red.

Thank you again for your comments. If our revised manuscript has any problem, please don’t hesitate to inform us. Your kind help will be my pleasure.
